# Factors associated with knowledge and practice regarding oxygen administration: A cross-sectional study among registered nurses working in wards and ICUs at Muhimbili National Hospital in Dar es Salaam, Tanzania

Magdalena S. Kimario[1], Joel Seme Ambikile [2]*, Masunga K. Iseselo[2]

1 Department of Surgical Intensive Care, Kilimanjaro Christian Medical Centre (KCMC), Kilimanjaro, Tanzania, 2 Clinical Nursing Department, Muhimbili University of Health and Allied Sciences (MUHAS), Dar es Salaam, Tanzania

* joelambikile@yahoo.com

## Abstract

### Background

Oxygen is a vital therapy approved by WHO, crucial for critically ill patients as a supplemental treatment. Nurses' pivotal role in oxygen administration is poorly understood. This study aimed to assess factors associated with oxygen administration to critically ill patients among nurses at Muhimbili National Hospital (MNH) in Dar es Salaam, Tanzania.

### Methods

A descriptive cross-sectional study was conducted among 208 nurses using a simple random technique to recruit respondents. Data were collected in May 2022 via a self-administered questionnaire, designed and tested by the authors. Mean knowledge and practice scores were used to categorize knowledge and practice as high or low respectively. The Chi-square test and multiple logistic regression analyses were performed to evaluate factors associated with knowledge and practice regarding oxygen administration. Statistical significance was determined at a $p$-value less than 0.05.

### Results

Of the 208 respondents, 96(46.2%) and 78(37.5%) had low knowledge and low practices regarding oxygen administration, respectively. Receiving in-service training on oxygen administration (AOR: 3.515; $p < 0.001$) was positively associated with knowledge of oxygen administration. None of the factors showed a statistically significant association with the practice of oxygen administration.

**Data Availability Statement:** All relevant data are within the manuscript and its Supporting Information files.

**Funding:** This research received funding from the Tanzanian government through the Ministry of Health The funders had no role in study design, data collection and analysis, decision to publish, or preparation of the manuscript.

**Competing interests:** The authors have declared that no competing interests exist.

## Conclusion

A substantial proportion of nurses had inadequate knowledge and practices regarding oxygen administration. While none of the assessed factors were found to have a statistical association with practice on oxygen administration, it is important to note their clinical significance. Healthcare institutions can benefit from implementing regular in-service training programs to address these knowledge and skills gaps and ensure that nurses are well-prepared for proper oxygen administration. Additionally, ongoing monitoring and support are essential to help translate improved knowledge into effective clinical practices.

## Introduction

Oxygen therapy is not just the most widely employed and cost-effective medicinal gas for critically ill patients: it is a lifeline that bridges the gap between life and death. Its indispensable role in saving lives has been dramatically highlighted in the wake of the COVID-19 pandemic, where the timely provision of supplemental oxygen was instrumental in the survival of millions [1]. Targeted at combating hypoxia and hypoxemia, two conditions marked by dangerously low oxygen levels in tissue and blood, oxygen therapy stands as a cornerstone intervention for patients facing critical health challenges [2–7]. The effective administration of this therapy has consistently a profound impact, correlating with significant reductions in mortality rates among vulnerable patient populations [8–11]. This underscores the urgent need for stakeholders in healthcare to prioritize and optimize oxygen therapy practices.

A standardized protocol for the safe application of therapeutic oxygen to patients is essential for improving both healthcare providers' performance and patient health outcomes [12]. While oxygen therapy plays a critical role in saving lives, its improper administration can result in either inadequate or excessive tissue oxygenation, both of which can adversely affect health outcomes in patients [13]. Therefore, it is crucial to safely administer oxygen therapy to critically ill patients, supported by a clear rationale for its use, the availability of oxygen delivery devices and indications for use, as well as the necessity for monitoring during the administration of oxygen therapy [14].

The pivotal role of nurses in administering oxygen therapy cannot be overstated [15], as they make up the majority of healthcare providers. Nurses must fully grasp the critical implications of oxygen administration, mastering oxygenation processes and pathophysiology, while skillfully balancing between hypoxia and hyperoxia to ensure patient survival [16]. Expertise in determining each patient's oxygen needs and administering it correctly is crucial for improving patient outcomes. Evidence shows that a deep understanding of oxygen therapy intricacies strongly correlates with proficient care in critically ill patients [17]. Adherence to best practices and standardized protocols is essential for nurses when providing oxygen therapy to critically ill patients. This guarantees optimal oxygen administration, maximizing its effectiveness in improving patient outcomes [12].

Evidence suggests significant variations in oxygen therapy practices among critical care nurses, highlighting the need for studies aimed at enhancing oxygen therapy for critically ill patients [18]. These disparities arise from differences in nurses' knowledge and skills across various settings. Beginning with knowledge, for instance, previous studies in Ethiopia have reported proportions of nurses with good knowledge ranging from 52 to 61.5% [19–21], while in Yemen and Egypt they were extremely low, i.e. 3% and 6%, respectively [22,23].

Contrastingly, much higher proportions have been reported in Pakistan (78.0%) [24]. Across these and other studies, predictors of knowledge regarding oxygen administration among nurses include level of education, availability of oxygen therapy guidelines or protocols, work experience, regular practice with oxygen, and training in supplemental oxygen therapy [25]. Regardless of the availability of the 2022 National Guideline on Oxygen Therapy in Tanzania [26], there is no existing evidence on nurses' knowledge of oxygen administration.

Similarly, studies examining oxygen administration practices in various settings have revealed variations in the proportions of nurses with such practices. In Ethiopia, previous studies have indicated levels of good practice on oxygen therapy ranging from 34.9% to 67.0% [20,27,28]. Factors influencing oxygen administration practices among nurses in critically ill patients include knowledge on oxygen therapy, work experience, workload, age, periodic maintenance and supply of equipment and devices, and labeling of the volume of the cylinder after use [19,24,25,27]. Furthermore, factors such as the work environment, training, availability of standard operating guidelines, managerial support, and power supply have been implicated in nurses' capacity to deliver effective oxygen therapy [11,28]. However, similar to knowledge, there is no existing evidence in Tanzania regarding nurses' practices of oxygen administration.

Studies reporting proportions of nurses with knowledge and practices regarding oxygen administration highlight an existing gap, indicative of key reasons for sub-optimal administration of oxygen therapy, particularly prevalent in low-income settings [17,29,30]. Examining context-specific factors associated with knowledge and practice regarding oxygen administration is crucial to bridge the existing gap. In Tanzania, for instance, a National Guideline on Oxygen Therapy was developed in 2022 in response to the COVID-19 pandemic, which exponentially increased the demand for oxygen. The guideline is intended for use by all healthcare professionals involved in providing oxygen and Essential Emergency and Critical Care, including ambulance staff, first responders, paramedics, doctors, nurses, midwives [26]. However, the implementation of this guideline requires evidence of the capacity of healthcare professionals to effectively and optimally administer oxygen therapy, which is currently lacking in the country. Thus, this study aimed to assess the levels of knowledge, practice and associated factors related to oxygen administration therapy among nurses caring for critically ill patients in Dar es Salaam, Tanzania. The findings from this study offer valuable insights into strategies for improving oxygen administration competencies among nurses, including regular training and informed policy decisions to enhance patient care and safety.

## Materials and methods

### Study setting

This study was conducted at Muhimbili National Hospital (MNH) in Dar es Salaam, Tanzania. Dar es Salaam is a very fast-growing and largest commercial city in Tanzania with a population of around 5.4 million [31,32]. MNH is the largest national referral hospital that provides services to patients from different parts of the country. The hospital was suitable for the study as it received referral patients mostly in need of advanced and specialized care from various parts of the country. It had a total of 1095 nurses, including 832 registered nurses (92 being graduates with various specialities and 263 enrolled nurses (holders of certificate qualification) [33]. Moreover, the hospital is equipped to offer a wide range of specialized services to local and international clients. At the time of the study, the hospital had a capacity of 1500 beds, attended between 1000 and 1200 outpatients per day, and admitted up to 1200 inpatients per week [34]. Furthermore, the hospital received patients with various health issues, many being in critical or serious condition that required a certain level of supplementary oxygen. Most of

these patients were admitted in the 4 main blocks of the hospital namely Kibasila, Maternity, Mwaisela, and Sewahaji.

## Study design

The study used a descriptive cross-sectional design using a quantitative approach. This approach was considered appropriate to quantify the factors associated with the knowledge and practice of oxygen administration therapy to critically ill patients among nurses at the same time [35,36].

## Population characteristics

The study population comprised registered nurses working in the four blocks constituting the medical, surgical, and obstetrics/gynecology wards at MNH, which encompassed ICUs. These nurses were the front liners in administering oxygen therapy to critically ill patients. The study included registered nurses (who had a diploma, bachelor's degree, or master's degree), with active practicing licenses. We excluded all nurses who were not available at work during the data collection period like those on study leave, acutely ill, and on long vacation.

## Sample size and sampling procedure

The sample size of 208 was calculated using a cross-sectional sample size formula for known population size, accounting for a 95% confidence level, a 5% margin of error, and an assumed proportion of nurses with sufficient knowledge of oxygen administration (p = 0.5) [37]. To achieve this sample size, a sampling frame of 360 nurses, provided by the administrative office, was used. The frame included nurses from the four hospital blocks: Kibasila (93 nurses), Maternity (83 nurses), Mwaisela (102 nurses), and Sewahaji (82 nurses). Proportional sampling was employed, resulting in the selection of 48 nurses from Kibasila, 45 from Maternity, 71 from Mwaisela, and 44 from Sewahaji.

All nurses in the sampling frame were initially informed about the study and invited to participate. The selection process involved random sampling in each block. For example, in Kibasila, 48 out of 93 pieces of paper were marked 'YES' (representing the required sample), while the remaining 45 were marked 'NO'. The papers were folded, placed in a box, thoroughly mixed, and each nurse selected one without replacement. Before the next nurse picked, the papers were mixed again. Nurses who drew a 'YES' were included in the study. This process was repeated for the other blocks to reach the total sample size of 208. The response rate was 100%, as all nurses in the sampling frame agreed to participate, and everyone who selected a 'YES' completed the questionnaire.

## Data collection tools and measurements

**Questionnaire.**  A self-administered questionnaire with close-ended questions of multiple choice and a Likert scale was used for data collection. The questionnaire was adopted from standard tools previously developed and used in a similar study in Cairo, Egypt [23], which involved being reviewed for their completeness and relevance by experts consisting of 3 professors from Medical Surgical Nursing specialty. The reported reliability for oxygen knowledge and practice were Cronbach's alphas of 0.87 and 0.89 respectively. The tools have also been previously used in another study in Pakistan [38]. The questionnaire comprised four sections including demographic characteristics (age, sex, marital status, level of professional education, work experience, in-service training on oxygen administration, and workstation), knowledge regarding oxygen administration, practice related to oxygen administration, and

organizational factors (**S1 Appendix**). The questionnaire was in English language and reflected various factors that might influence the administration of oxygen therapy among nurses. In addition, the questionnaire was pre-tested at the Mloganzira site (which was part of MNH at a different location in the same city) with 21 nurses, representing 10% of the sample size used in a previous study [39]. Feedback indicated that the questionnaire was relevant.

**Measurements.** As explanatory variables, sociodemographic and organizational characteristics were scored as categorical variables. These were determined if a specific criterion was fulfilled or not, or either present or absent.

As an outcome variable, knowledge regarding the administration of oxygen therapy was assessed using 11 items with multiple-choice questions. As in a previous study, the knowledge percentage score for each respondent was calculated by summing up the number of correct responses divided by the total number of items assessed (11), multiplied by 100%. [20]. Respondents' mean knowledge score was used as a cut-off point to categorize knowledge as either high or low, i.e. any score equal to or above the mean knowledge was considered as high and below it as low.

To assess oxygen administration practice as the secondary outcome variable, 27 items, each consisting of 4-point Likert scale responses ranging from 1 to 4 which represented never, sometimes, often, and always, respectively, were used. Therefore, the minimum score was 27, and the maximum was 108. The mean respondents' score was used to categorize practice as either high or low, i.e. high practice was considered when the score was equal to or above the respondents' mean score and low practice when it was below.

## Data collection procedure

The process of recruiting respondents for this study began on 11[th] May and ended on 27[th] May 2022. Data were collected at the end of the shifts when nurses were about to go home. Nurses were approached in each ward and ICU and requested to fill the questionnaires in at the end of shifts with assistance from the nurse in charge. Information about the study including objectives was provided to potential respondents and those who fulfilled inclusion criteria were asked to fill out the questionnaires after obtaining written consent from them. Filling out the questionnaires was carried out in a convenient room within or close to their working stations. Two research assistants who had experience in nursing research and good communication skills supervised the data collection activity. They were trained for two days on the study aims, methodology, data collection, and sampling techniques. Moreover, the use of different strategies in data collection to maximize the response rate was emphasized including being good-mannered, anonymity, and observing confidentiality during the administration and collection of questionnaires.

## Validity and reliability

The questionnaire was reviewed by an expert in the nursing field to check for the content validity of the tool. The experts were requested to review each question to ascertain if it answered the research objectives. The feedback from the experts was critically reviewed to improve the content of each question item. No major changes were made after the expert review. To enhance reliability, the questionnaire was pre-tested among respondents with similar characteristics to our study population, who were not involved in the actual study as previously explained [40]. This ensured the data collection tools provided accurate and reliable data for the study. Therefore, after adjustment, the same tool was administered to all participants by researchers. Cronbach's alpha coefficient of 0.78 was obtained as a statistical measure of reliability for the Likert scale questions used to assess oxygen administration practice.

## Data management and analysis

Thorough data cleaning and validation checks were performed to identify and address any inconsistencies or errors in the data. Data from the questionnaires were first entered into an Excel worksheet before being transferred into the software used for analysis. The IBM Statistical Package for Social Sciences (SPSS) version 25 software was used to analyze data. Continuous variables were summarized using means and standard deviations while categorical variables were summarized using frequencies and percentages. The association between independent and dependent variables was determined by performing the Chi-square test and multiple logistic regression analyses. Variables with p-values < 0.2 from the Chi-square test were selected and included in the multiple logistic regression model. All variables with a P-value ≤ 0.05 and a 95% confidence level were considered statistically significant.

## Ethical consideration

Ethical approval was obtained from the Research Ethics Committee of Muhimbili University of Health and Allied Sciences (MUHAS) with IRB No. MUHAS-REC-04-2022-1123 and letter Reference No. DA282/298/01.C/1123. Permission for data collection was obtained from the Muhimbili National Hospital administration. Written informed consent was obtained from all respondents before they participated in the study. Respondents were given information about the aim, nature, benefit, and risk of the study and were requested to participate in the study after ensuring the information provided was understood. Moreover, the right to withdraw from the study was clearly described and that participation was voluntary. Confidentiality was ensured through the anonymity of all documents that contained participants' information.

# Results

## Socio-demographic characteristics of the respondents

Among 208 nurses who participated in this study, their mean age (SD) was 36.0(9.2), 120 (57.7%) were female, and about two-thirds (144; 69.2%) were in the 20–40 age group. More than half (113; 54.3%) were married, 107 (51.4%) had a diploma in nursing, and half (104; 50.0%) had one to five years of working experience. A hundred and seventy-five (84.1%) worked in the wards (as opposed to ICU), and 113 (54.3%) did not receive any in-service oxygen therapy administering training (Table 1).

## Nurses' knowledge of oxygen administration to critically ill patients

The mean (SD) knowledge score regarding oxygen administration to critically ill patients was 6.6 (2.1), and 96 (46.2%) respondents had low knowledge. Items with the highest knowledge score were the unit of oxygen administration (186; 89.4%) followed by two items i.e. approximated oxygen concentration delivered by nasal cannula and the problem associated with simple face mask items, both with 146 (70.2%). Items with the lowest knowledge score were contraindications to oxygen administration and precautions to be taken during the administration of oxygen therapy, with 151 (72.6%) and 124 (59.6%) scores, respectively.

## Factors associated with nurses' knowledge of oxygen administration

A univariate analysis using the Chi-square test ($X^2$) was performed to assess sociodemographic factors associated with nurses' knowledge of oxygen administration (Table 2). Higher professional education ($X^2 = 5.748$; $p = 0.017$), shorter work experience ($X^2 = 4.952$; $p = 0.027$), and receiving in-service training on oxygen administration ($X^2 = 22.126$; $p<0.001$) showed significant associations with nurses' knowledge of oxygen administration.

**Table 1. Nurses' socio-demographic characteristics (N = 208).**

| Variable | Categories | Frequency | Percentages (%) |
|---|---|---|---|
| Age (Years) | 20–40 | 144 | 69.2 |
|  | 41–60 | 64 | 30.8 |
| Sex | Female | 120 | 57.7 |
|  | Male | 88 | 42.3 |
| Marital status | Single | 95 | 45.7 |
|  | Married | 113 | 54.3 |
| Level of professional education | Diploma | 107 | 51.4 |
|  | Degree/Masters | 101 | 48.6 |
| Work experience | 1 to 5 years | 104 | 50.0 |
|  | >5 years | 104 | 50.0 |
| Inservice trained on $O_2$ admin | No | 113 | 54.3 |
|  | Yes | 95 | 45.7 |
| Work section | Ward | 175 | 84.1 |
|  | ICU | 33 | 15.9 |

Further multivariate analysis was performed using multiple logistic regression, with inclusion in the model of sociodemographic factors with $p$ values of $\leq 0.2$. This was done to exclude variables with $p$ values very far from 0.05 (Table 3). Only receiving in-service training on oxygen administration (AOR: 0.285; CI: 0.157, 0.519; $p<0.001$) remained statistically significant. The Hosmer-Lemeshow goodness-of-fit test yielded a Chi-square of 2.840, 7 degrees of freedom, and a *p-value* of 0.899, indicating no evidence of lack of goodness-of-fit.

## Nurses' reported practice regarding oxygen administration to critically ill patients

The mean (SD) practice score regarding oxygen administration to critically ill patients was 89.2 (12.6), and 78 (37.5%) respondents had low practice. Items with the highest practice score

**Table 2. Univariate analysis of factors associated with nurses' knowledge of oxygen administration (N = 208).**

| Variable | Categories | Low knowledge (mean < 6.6) | | High knowledge (mean ≥ 6.6) | | $X^2$ | p-value |
|---|---|---|---|---|---|---|---|
|  |  | n | % | n | % |  |  |
| Age (Years) | 20–40 | 63 | 43.8 | 81 | 56.3 | 1.088 | 0.297 |
|  | 41–60 (ref.) | 33 | 51.6 | 31 | 48.4 |  |  |
| Sex | Female | 55 | 45.8 | 65 | 54.2 | 0.012 | 0.914 |
|  | Male (ref.) | 41 | 46.6 | 47 | 53.4 |  |  |
| Marital status | Single | 42 | 44.2 | 53 | 55.8 | 0.266 | 0.606 |
|  | Married (ref.) | 54 | 47.8 | 59 | 52.2 |  |  |
| Levels of professional Education. | Diploma | 58 | 54.2 | 49 | 45.8 | 5.748 | 0.017 |
|  | Degree/Masters (ref.) | 38 | 37.6 | 63 | 62.4 |  |  |
| Work experience | 1 to 5 years | 40 | 38.5 | 64 | 61.5 | 4.952 | 0.026 |
|  | >5 years (ref.) | 56 | 53.8 | 48 | 46.2 |  |  |
| Inservice training | No | 69 | 61.1 | 44 | 38.9 | 22.126 | <0.001 |
|  | Yes (ref.) | 27 | 28.4 | 68 | 71.6 |  |  |
| Work section | Ward | 85 | 48.6 | 90 | 51.4 | 2.594 | 0.107 |
|  | ICU (ref.) | 11 | 33.3 | 22 | 66.7 |  |  |

A Chi-square test was performed.

**Table 3. Multivariate analysis of factors associated with nurses' knowledge of oxygen administration (N = 208).**

| Variable | Categories | Low knowledge (mean < 6.6) | | High knowledge (mean ≥ 6.6) | | Multiple logistic regression | | |
|---|---|---|---|---|---|---|---|---|
| | | n | % | n | % | AOR | 95% CI | p-value |
| Levels of professional Education. | Diploma | 58 | 54.2 | 49 | 45.8 | 0.557 | 0.308, 1.007 | 0.053 |
| | Degree/Masters (ref.) | 38 | 37.6 | 63 | 62.4 | | | |
| Work experience | 1 to 5 years | 40 | 38.5 | 64 | 61.5 | 1.757 | 0.973, 3.174 | 0.062 |
| | >5 years (ref.) | 56 | 53.8 | 48 | 46.2 | | | |
| Inservice training | No | 69 | 61.1 | 44 | 38.9 | 0.285 | 0.157, 0.519 | <0.001 |
| | Yes (ref.) | 27 | 28.4 | 68 | 71.6 | | | |
| Work section | Ward | 85 | 48.6 | 90 | 51.4 | 0.639 | 0.281, 1.451 | 0.284 |
| | ICU (ref.) | 11 | 33.3 | 22 | 66.7 | | | |

Multiple logistic regression analyses were performed. Key: AOR; Adjusted odds ratio, CI; Confidence interval.

were ensuring pulse oximetry to monitor response to oxygen therapy was available before administration of oxygen and removing gloves after administration of oxygen with both scores being 195 (93.8%). Items with the lowest practice score were discarding used equipment after administration (137; 65.9%) and verifying physician prescriptions before administration (122; 58.7%). Part of the tool measuring oxygen administration practice was tested for its internal consistency reliability, with a Cronbach's alpha of 0.947 for all 27 items.

### Factors associated with nurses' practice of oxygen administration

The univariate analysis using the Chi-square test was performed to assess sociodemographic, knowledge, and organizational factors associated with nurses' practice regarding oxygen administration (Table 4). None of these factors was associated with the practice of oxygen administration.

Further multivariate analysis was performed using multiple logistic regression, with inclusion in the model of sociodemographic factors with p values of ≤ 0.2, to exclude variables with $p$ values very far from 0.05 (Table 5). Still, none of these factors was associated with the practice of oxygen administration. The Hosmer-Lemeshow goodness-of-fit test yielded a Chi-square of 6.241, 8 degrees of freedom, and a $p$-value of 0.620, indicating no evidence of lack of goodness-of-fit.

### Discussion

This study assessed factors influencing nurses' knowledge and practice in administering oxygen therapy to critically ill patients. The findings revealed that nurses who received in-service training on oxygen administration demonstrated significantly higher levels of knowledge in this area. However, no significant associations were found between socio-demographic or organizational factors and nurses' practice of oxygen administration. These results suggest that while targeted training enhances theoretical knowledge, other variables may need to be explored to improve the practical aspects of oxygen administration.

Receiving in-service training in our study was associated with three-and-a-half-fold increase in the likelihood of nurses possessing high knowledge of oxygen administration. This strongly supports the efficacy of on-the-job training as a key method for improving nurses' understanding of oxygen administration, consistent with similar findings from a study in Ethiopia [28]. Additionally, studies have documented that targeted training programs significantly elevates nurses' knowledge levels in this area [41–43]. Given this robust evidence, it is critical

**Table 4. Univariate analysis of factors associated with Nurses' practice on oxygen administration (N = 208).**

| Variable | Categories | Low practice (mean <89.2%) | | High practice (mean ≥89.2%) | | $X^2$ | p-value |
|---|---|---|---|---|---|---|---|
| | | n | % | n | % | | |
| Age (Years) | 20–40 | 51 | 35.4 | 93 | 64.6 | 0.867 | 0.352 |
| | 41–60 (ref.) | 27 | 42.2 | 37 | 57.8 | | |
| Sex | Female | 45 | 37.5 | 75 | 62.5 | 0.000 | 1.000 |
| | Male (ref.) | 33 | 37.5 | 55 | 62.5 | | |
| Marital status | Single | 37 | 38.9 | 58 | 61.1 | 0.156 | 0.693 |
| | Married (ref.) | 41 | 36.3 | 72 | 63.7 | | |
| Level of prof. Edu. | Diploma | 38 | 35.5 | 69 | 64.5 | 0.371 | 0.543 |
| | Degree/Masters (ref.) | 40 | 39.6 | 61 | 60.4 | | |
| Work experience | 1 to 5 years | 34 | 32.7 | 70 | 67.3 | 2.051 | 0.152 |
| | >5 years (ref.) | 44 | 42.3 | 60 | 57.7 | | |
| Inservice training | No | 46 | 40.7 | 67 | 59.3 | 1.086 | 0.297 |
| | Yes (ref.) | 32 | 33.7 | 63 | 66.3 | | |
| Work section | Ward | 70 | 40.0 | 105 | 60.0 | 2.941 | 0.086 |
| | ICU (ref.) | 8 | 24.2 | 25 | 75.8 | | |
| Knowledge of $O_2$ admin. | Low | 42 | 43.8 | 54 | 56.3 | 2.971 | 0.085 |
| | High (ref.) | 36 | 32.1 | 76 | 67.9 | | |
| Available training courses | No | 71 | 37.8 | 117 | 62.2 | 0.059 | 0.808 |
| | Yes (ref.) | 7 | 35.0 | 13 | 65.0 | | |
| Availability of equip/supplies | No | 44 | 32.8 | 90 | 67.2 | 3.496 | 0.062 |
| | Yes (ref.) | 34 | 45.9 | 40 | 54.1 | | |
| Periodic equip./devices maint. | No | 55 | 35.7 | 99 | 64.3 | 0.807 | 0.369 |
| | Yes (ref.) | 23 | 42.6 | 31 | 57.4 | | |
| Availability STD protocol | No | 61 | 35.9 | 109 | 64.1 | 1.039 | 0.308 |
| | Yes (ref.) | 17 | 44.7 | 21 | 55.3 | | |
| Clear/complete prescriptions | No | 47 | 34.1 | 91 | 65.9 | 2.073 | 0.150 |
| | Yes (ref.) | 31 | 44.3 | 39 | 55.7 | | |
| Oral prescription only | No | 46 | 33.8 | 90 | 66.2 | 2.266 | 0.132 |
| | Yes (ref.) | 32 | 44.4 | 40 | 55.6 | | |

A Chi-square test was performed.

to prioritize in-service training initiatives to ensure that nurses are well-equipped to administer oxygen therapy effectively. Thus, prioritizing training, alongside other evidence-based strategies, is essential for optimizing nurses' knowledge and improving patient outcomes [44].

Our study did not reveal a statistically significant association between the level of professional education among nurses and their knowledge of oxygen administration. This suggests that nurses with diploma qualifications had comparable knowledge to those with degree or master's qualifications in this area. These findings are consistent with research from Iraq and Pakistan [24,45], which similarly found no significant differences based on educational level. However, our results contradict a study in Turkey, where nurses with higher levels of education demonstrated better knowledge of oxygen administration [46]. This discrepancy may be explained by various factors, such as differences in clinical experience, the specific nursing environment, critical thinking skills, and the depth of curricula across different educational programs. Other influences, such as continuing educational opportunities, scope of practice,

**Table 5. Multivariate analysis of factors associated with Nurses' practice on oxygen administration (N = 208).**

| Variable | Categories | Low practice (mean <89.2%) | | High practice (mean ≥89.2%) | | Multiple logistic regression | | |
|---|---|---|---|---|---|---|---|---|
| | | n | % | n | % | AOR | 95%CI | *p*-value |
| Work experience | 1 to 5 years | 34 | 32.7 | 70 | 67.3 | 1.343 | 0.748, 2.412 | 0.324 |
| | >5 years (ref.) | 44 | 42.3 | 60 | 57.7 | | | |
| Work section | Ward | 70 | 40.0 | 105 | 60.0 | 0.473 | 0.198, 1.134 | 0.093 |
| | ICU (ref.) | 8 | 24.2 | 25 | 75.8 | | | |
| Knowledge of O₂ admin. | Low | 42 | 43.8 | 54 | 56.3 | 0.642 | 0.357, 1.156 | 0.140 |
| | High (ref.) | 36 | 32.1 | 76 | 67.9 | | | |
| Availability of equip/supplies | No | 44 | 32.8 | 90 | 67.2 | 1.665 | 0.869, 3.190 | 0.125 |
| | Yes (ref.) | 34 | 45.9 | 40 | 54.1 | | | |
| Clear/complete prescriptions | No | 47 | 34.1 | 91 | 65.9 | 1.286 | 0.601, 2.753 | 0.517 |
| | Yes (ref.) | 31 | 44.3 | 39 | 55.7 | | | |
| Oral prescription only | No | 46 | 33.8 | 90 | 66.2 | 1.129 | 0.510, 2.495 | 0.765 |
| | Yes (ref.) | 32 | 44.4 | 40 | 55.6 | | | |

Multiple logistic regression analyses were performed. Key: AOR; Adjusted odds ratio, CI; Confidence interval.

professional autonomy, and individual personal factors, may also play a role in shaping nurses' understanding of oxygen administration [47]. While the lack of significant differences in our study might suggest that on-the-job training are more crucial than formal education in this context, it remains important to consider how this and other factors might interact to impact overall knowledge. Future research should explore how these elements combine to influence nurses' practical knowledge and application of oxygen therapy.

The current study found no statistically significant association between years of work experience and knowledge of oxygen administration. This suggests that simply having more years in the field does not guarantee a deeper understanding of this critical skill. Supporting our findings, a similar study conducted in Iran also reported no significant relationship between work experience and knowledge of oxygen administration [48]. Contrastingly, research fromn Ethiopia indicated that less work experience was associated with a better understanding of oxygen administration [25]. Furthermore, several studies from both Ethiopia and Turkey have identified a positive correlation between longer work experience and increased knowledge of oxygen administration [44,46,49,50]. These varying results highlight that factors beyond mere years of experience play a crucial role in shaping clinical competency. Elements such as the workplace environment, educational background, clinical training, job satisfaction, turnover intention, job stress, and critical thinking, are all significant contributors to knowledge acquisition and overall clinical effectiveness [51]. This complexity underscores the need for a multi-faceted approach to training and professional development in healthcare settings.

While our study did not find a predictive relationship between knowledge of oxygen administration and practical application, research in Ethiopia shown that nurses with a strong understanding of supplemental oxygen administration are more likely to demonstrate proficient practices compared to those with less knowledge [28,49]. Additionally, a systematic review underscores the essential role of theoretical knowledge in developing clinical competency. It also points to other influencial factors, including workplace environment, job satisfaction, turnover intention, job stress, and critical thinking, which may contribute to the observed differences in oxygen administration practices [51]. This suggests that enhancing both knowledge and the surrounding conditions could be vital for improving clinical outcomes in oxygen administration.

In our study, none of the assessed factors were found to predict practice in oxygen adminis-tration. This contrasts with findings from other research, which present a mixed picture. For instance, similar to our study, a study in Egypt indicated that the age of participants had no sig-nificant association with their practice of administering oxygen therapy [23]. However, unlike our findings, studies in Ethiopia and Denmark identified that a higher level of professional education, receiving specific training, and working in an ICU, were significant predictors of effective oxygen administration practices [28,52]. These differences may stem from various factors previously mentioned, such as workplace environment, job satisfaction, turnover intention, job stress, and critical thinking [51]. This highlights the complexity of the relation-ship between knowledge and practice, suggesting that multiple contextual elements must be considered to understand clinical competencies fully.

In our study, organizational factors did not show a significant impact on the levels of oxy-gen administration practice. However, other research has identified various organizational ele-ments that do influence practice. For isntance, studies in Saudi Arabia and Ghana found that factors such as workload, availability of local guidelines, adherence to standardized protocols, and the accessibility and cost of delivery devices and oxygen supply were associated with effec-tive oxygen practices [12,53,54]. Additonally, other studies have pointed out that issues such as inadequate supply of oxygen equipment, poor maintenance of the oxygen system, and a lack of oxygen prescriptions from doctors significantly contributes to low levels of practice among nurses administering oxygen therapy to critically ill patients [23,55]. These findings suggest that the factors affecting oxygen administration practice are multifaceted, highliting the need for or a comprehensive approach that addresses both individual and organizational challenges to improve clinical practices.

The results of this study may have some methodological limitations based on a few issues. First, we used a self-reported assessment regarding oxygen administration practice rather than observing the actual practice, which may not reflect a real situation. Second, social desirability could have affected results since the respondents' responses might reflect their perception of desired responses rather than the true responses toward the practice of administering oxygen therapy to critically ill patients. The two limitations might have influenced scores, especially in favor of respondents with higher knowledge and practice scores. Third, being cross-sectional, the study cannot ascertain a causal-effect relationship between sociodemographic and organi-zational factors and knowledge and practice regarding oxygen administration. To mitigate the limitations, respondents were assured of anonymity and confidentiality to enhance true responses from them. However, results from this study highlight important factors that may affect the administration of oxygen to critically ill patients among nurses in health facilities.

In conclusion, a substantial proportion of nurses in this study had inadequate knowledge and practice regarding oxygen administration. While receiving in-service training was posi-tively associated with knowledge, none of the demographic and organizational factors were significant predictors of practice regarding oxygen administration. However, it is important to note the clinical significance these factors. Healthcare institutions can benefit from imple-menting regular in-service training programs to address the knowledge and skills gaps and ensure that nurses are well-prepared for proper oxygen administration. Additionally, ongoing monitoring and support are essential to help translate improved knowledge into effective clini-cal practices."

## Clinical implications and scientific contribution of the study

This study highlights the urgent need for targeted nurse education to address gaps in knowl-edge and practice, reducing errors in oxygen administration. To improve patient outcomes

and safety, hospitals should implement mandatory, regular in-service training on oxygen therapy for nurses, particularly in critical care settings. These findings can inform similar interventions in other low-resource settings, potentially improving global standards in oxygen therapy.

## Supporting information

**S1 Appendix. Items assessed in the questionnaire covering knowledge, practice, and organizational factors.**
(PDF)

## Acknowledgments

We would like to thank the Muhimbili National Hospital for permitting us to conduct this study. We specifically appreciate the time and valuable information provided by nurses who worked in various wards and intensive care units in the hospital that made this study a success.

## Author Contributions

**Conceptualization:** Magdalena S. Kimario.

**Data curation:** Magdalena S. Kimario.

**Formal analysis:** Joel Seme Ambikile, Masunga K. Iseselo.

**Funding acquisition:** Magdalena S. Kimario.

**Investigation:** Magdalena S. Kimario.

**Methodology:** Magdalena S. Kimario, Joel Seme Ambikile, Masunga K. Iseselo.

**Project administration:** Joel Seme Ambikile.

**Resources:** Magdalena S. Kimario.

**Software:** Joel Seme Ambikile.

**Supervision:** Joel Seme Ambikile, Masunga K. Iseselo.

**Writing – original draft:** Magdalena S. Kimario.

**Writing – review & editing:** Magdalena S. Kimario, Joel Seme Ambikile, Masunga K. Iseselo.

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
