## [Decision Letter · Decision Letter 0]

2 Feb 2024

PONE-D-23-37459Factors associated with knowledge and practice regarding oxygen administration: A cross-sectional study among registered nurses in Dar es Salaam, TanzaniaPLOS ONE

Dear Dr. Ambikile,

Thank you for submitting your manuscript to PLOS ONE. After careful consideration, we feel that it has merit but does not fully meet PLOS ONE’s publication criteria as it currently stands. Therefore, we invite you to submit a revised version of the manuscript that addresses the points raised during the review process.

We look forward to receiving your revised manuscript.

Kind regards,

Agegnehu Bante Getenet

Academic Editor

PLOS ONE

Journal Requirements:

This research received funding from the Tanzanian government through the Ministry of Health

Additional Editor Comments:

Critical evaluation of the manuscript entitled: Factors associated with knowledge and practice regarding oxygen administration: A cross-sectional study among registered nurses in Dar es Salaam, Tanzania

Abstract

Limit the amount of information in the background section of the abstract. On the other hand, the method section misses key information such as the study period, tool, and method of data entry.

In the result section, remove unnecessary detail in reporting the background characteristics of the participants. Include the 95% CI when you report the outcome variables. Remove either the p-value or the CI from the factors; it transmits the same message.

Introduction

A paragraph must describe a single idea; the breakdown paragraph is at line 51. A detailed explanation of hypoxia and hypoxemia is not necessary. Above all, the background lacks the proportion of knowledge and practice among nurses from the previous studies. Address the factors associated with knowledge and practice separately.

Support the descriptions with figures; for example, in line 73, you write, “In Tanzania, poor administration of oxygen therapy is a growing problem." What is your evidence? Are there any proposed solutions so far?

Methods and materials

Modify the order of the subsections as suggested by one of the reviewers. The study setting must be written in relation to your title. Since your study is on nurses, you have to mention the total number of nurses working in the hospital.

Exclusion criteria: you exclude those nurses with less than one year of experience caring for patients in need of oxygen therapy. On the other hand, experience is considered a variable. How?

Why the correction formula? It is better if you include all of them. Some of the problems observed in the model are due to the small sample size. The approach used to select the participants is not scientific. Why do you not use a table of random numbers in Excel by using the nurse's ID? Did you really write on 360 pieces of paper—208 yes and no for the rest????

Modify the subheadings under the method section; e.g., “Replace “’Data collection’’ with ‘’data collection procedure’’

How did you minimize bias during the data collection?

Nothing is written about the software used for the data entry. Which version of SPSS was used? 23 vs. 25. Be consistent. The criterion to include variables for the multiple logistic regression must be mentioned here.

Result

Table 2: Why knowledge and practice are presented with the socio-demographic.

Table 3. Add the cross-tabulation for each variable (the poor category must be included). Moreover, the operational definition of good vs. poor knowledge is not clear. Under the method section, you write that the mean is used as a cutoff point. In contrast, in this table, you put proportion. How? The mean you put in lines 202 is not clear. Knowledge was assessed by 11 items; how is the mean 60?

Table 4. Run the multiple logistic regression even if there is no significant variable in the bivariable analysis.

Discussion

The 95% confidence interval is mandatory to compare your finding with the previous studies.

The justifications given for the discrepancies are very shallow and not supported by evidence.

Add the clinical implications and scientific contribution of this study.

In general, the manuscript needs further reanalysis and revision of all its components.

Reviewers' comments:

Reviewer's Responses to Questions

**Comments to the Author**

1. Is the manuscript technically sound, and do the data support the conclusions?

Reviewer #1: Yes

Reviewer #2: Yes

2. Has the statistical analysis been performed appropriately and rigorously? 

Reviewer #1: Yes

Reviewer #2: No

3. Have the authors made all data underlying the findings in their manuscript fully available?

Reviewer #1: Yes

Reviewer #2: Yes

4. Is the manuscript presented in an intelligible fashion and written in standard English?

Reviewer #1: Yes

Reviewer #2: Yes

5. Review Comments to the Author

Reviewer #1: General comment

Overall, the manuscript is well written, and it addresses important issue “Knowledge and practice regarding oxygen administration” in low-resource countries like Tanzania. The problem has been well justified with the support of extensive literature review. The methods and results are well presented. However, I have few comments that would help to further improve the manuscript:

Major comments

1. Abstract: Briefly describe the measurement of outcome variable in the abstract section of the manuscript

2. Validity and reliability: I recommend reporting the statistical measures of reliability for Likert scale questions (e.g Cronbach alpha)

3. Multiple logistic regression: I would recommend setting the criteria to select variables to include in the logistic regression model and describing it in the methods section. My thinking is that the model would have been improved by excluding sex and marital status because their p-values are very far from 0.05.

Minor comments

4. Line 27: Consider rephrasing “A p-value of less than 0.05 was used to determine significant relationships” to “the p-value less than 0.05 was considered statistically significant”. This is because p-value less than 0.05 is just a criterion but not statistical method itself. The method used here was logistic regression.

5. Line 37 and line 324: I would recommend interpreting proportion instead of absolute number in the conclusion.

6. Line 39-541: The statement “Further research is needed to determine the relationship between length of work experience and knowledge of oxygen administration” Contradict with the presented results where authors report that experience was significantly associated with knowledge of oxygen administration. Consider rephrasing it.

7. Line 82: Rephrase “A descriptive, cross-sectional …” to “ A descriptive cross-sectional…”. I think Comma (,) is misplaced between the term “descriptive” and “cross-sectional”.

8. Line 92: “According to the hospital website”. The website should be appropriately cited as per recommended journal style.

9. Line 107: Consider rephrasing the statement “A sample size was calculated…” to “a sample size was estimated using the cross-sectional formula for study when the population size is known”.

10. Line 108: For readability purpose, the formula should not be presented inside the paragraph containing texts.

11. Line 110: I would recommend motivating the use of 10% as a non-response rate. Was it based on literature or something else?

12. Line 136-148 (Measurements): I would recommend you organize into paragraphs (if possible, use sub-titles). One or two paragraphs for of outcome variables (knowledge, practice) and one paragraph for explanatory variables (demographics, organizational, etc).

13. Line 173: the version of statistical package differs from the one stated in the abstract. Please revise to be consistent on the version of the statistical package used.

14. Line 175-176: “Descriptive data for continuous variables were analyzed using a measure of central tendencies such as means and standard deviation”. I think authors need just to state specifically the summary measures used (e.g Continuous variables were summarized using means and standard deviations).

15. Line 176-177: “The sociodemographic characteristics, nurses’ knowledge, nurses’ practice, and organization factors were analyzed using frequencies and percentages”. I think it is not clear why frequency and percentages. The reason is because they are categorical variables. Therfore, I would recommend rephrasing to a “categorical variables were summarized using frequencies and percentages”.

16. Line 179: “All variables with a p-value ≤ 0.05 and a 95% confidence level were considered statistically significant.” is not clear. Consider rephrasing to “All variables with p-value<0.05 (strictly less than 0.05 as in the abstract) was considered significant”. There is no need to use both significance level and confidence level as they mean the same.

17. Table 4: Indicate the reference category.

Reviewer #2: I would like say thank you for giving the chance to review this manuscript.

Title

It is a good title that assess the HCWs in the health facilities and support the different stakeholders.

Generally, it is good. However, the following comments/questions need to be considered before publication of this manuscript.

Abstracts

You include all sub-section of abstract, which is good. It is better if you remove less important finding from results subsection, for better attraction of readers.

-‘’ Of the 208 respondents, 120(57.7%) were female, 144(69.2%) were in the age group of 20-40, and 175(84.1%) worked in the wards”

- ‘’ None of the socio-demographic (including knowledge of oxygen administration)

and organizational factors were associated with practice on oxygen administration”

Materials and Methods

- You start with study design. It is better if begin from study setting since most of the articles are like that.

- You didn’t mention your study period.

- You specifically mention study population, again it is advisable to replace by population/participants characteristics since it include study criteria.

- Under data management and analysis, which type of SPSS Version you have been used specifically? You wrote two version, be consistent. 23/25?

- What cut of point you have been used during bivariate logistic regression to select the candidate of multivariate ?

- You didn’t mention nothing about your data entry mechanism? Model fitness?

Results

- What was the mean age of your respondents ?

- Table 3? It is not clear for the readers? Try to modify as per standard of regression table. Knowledge status (Good vs Poor; No of/frequency of single among good knowledge vs poor knowledge)? How some body know whether your COR calculation is correct or not? It is better if you modify it.

- Table 4 ? How you identify candidate variables ? by p-value <0.2/from previous articles/scientific consideration ? Did you consider the reason why not statistically significant but clinically significant ?

Discussion

- What is your standard reference to say lower or higher proportion ? I mean you didn’t incorporate CI for your both knowledge and practice ? Include it.

6. PLOS authors have the option to publish the peer review history of their article (what does this mean?). If published, this will include your full peer review and any attached files.

Reviewer #1: **Yes: **Christopher Mbotwa

Reviewer #2: No

---

## [Author Response · Author response to Decision Letter 0]

29 Feb 2024

We have responded to spefic reviewer and editor comments in the attached file labelled Response to Reviwers comments

---

## [Decision Letter · Decision Letter 1]

1 May 2024

PONE-D-23-37459R1Factors associated with knowledge and practice regarding oxygen administration: A cross-sectional study among registered nurses at Muhimbili National Hospital in Dar es Salaam, TanzaniaPLOS ONE

Dear Dr. Ambikile,

Thank you for submitting your manuscript to PLOS ONE. After careful consideration, we feel that it has merit but does not fully meet PLOS ONE’s publication criteria as it currently stands. Therefore, we invite you to submit a revised version of the manuscript that addresses the points raised during the review process.

We look forward to receiving your revised manuscript.

Kind regards,

Agegnehu Bante Getenet

Academic Editor

PLOS ONE

Additional Editor Comments:

The manuscript still needs further major revision; intensively address all the comments raised by the reviewers.

Reviewers' comments:

Reviewer's Responses to Questions

**Comments to the Author**

1. If the authors have adequately addressed your comments raised in a previous round of review and you feel that this manuscript is now acceptable for publication, you may indicate that here to bypass the “Comments to the Author” section, enter your conflict of interest statement in the “Confidential to Editor” section, and submit your "Accept" recommendation.

Reviewer #2: (No Response)

Reviewer #3: All comments have been addressed

2. Is the manuscript technically sound, and do the data support the conclusions?

Reviewer #2: Yes

Reviewer #3: Partly

3. Has the statistical analysis been performed appropriately and rigorously? 

Reviewer #2: No

Reviewer #3: Yes

4. Have the authors made all data underlying the findings in their manuscript fully available?

Reviewer #2: Yes

Reviewer #3: No

5. Is the manuscript presented in an intelligible fashion and written in standard English?

Reviewer #2: Yes

Reviewer #3: Yes

6. Review Comments to the Author

Reviewer #2: (No Response)

Reviewer #3: Title: Factors associated with knowledge and practice regarding oxygen administration: A cross-sectional study among registered nurses in Dar es Salaam, Tanzania

Version 1: Date 26/4/2024

Review reports:

Overall, it is well described paper that authors attempted give insights on” Factors associated with knowledge and practice regarding oxygen administration: A cross-sectional study among registered nurses in Dar es Salaam, Tanzania” However, I have given the following suggestions under each heading of the manuscript.

Title: It is interesting and relevant to the field that will be input for literature for future researchers. But study population looks general. Thus, it is a good idea to make it specific like nursing working at ED, ICU, ward and etc.

Abstract: Well written summarizing what was done and what was found.However,its result part regarding associations should described as positively or negatively associated. Additionally, I will suggest you refer back whether it is academically appropriate to say poor knowledge rather than low/high knowledge on something.

Introduction:

It clearly summarizes the current state of the topic and addresses the limitations of current knowledge in this field. Additionally, it clearly explains why the study was necessary.

Material and methods:

The study design and methods appropriate for the research question. The timeframe of the study sufficient is to see outcomes. It is clear how samples were collected or how participants were recruited from each unit. I am curious what was the sampling frame that authors used for simple random sampling technique and why authors didn’t take 360 nurses as it was feasible. Additionally, I have found one concern under measurements section, authors used mean score to classify knowledge as poor when score is less than mean and good when score greater than is mean. What about those who scored exactly mean? Similar question for practice?

Results:

The results are presented clearly and accurately, and all the relevant data have been included. But the data described in the text inconsistent with the data in the tables 2 where o2 admin knowledge and practice were described under the Table2 heading “nurses’ Socio-demographic characteristics.”

Discussion:

I applaud that author logically explained the findings and compared the findings with current findings in the research field and discussed contradictory data.However,samplze size difference as explanation does not sound thus it is advisable to look other scientific justifications for discrepancies. Additionally, the implications of the findings for future research and potential applications as well as limitations of the study are discussed.

Conclusion:

Authors provided a clear summary of the main points.

7. PLOS authors have the option to publish the peer review history of their article (what does this mean?). If published, this will include your full peer review and any attached files.

Reviewer #2: No

Reviewer #3: No

---

## [Author Response · Author response to Decision Letter 1]

25 Jun 2024

We hav gone through the comments point by point and have addressed all the comments raised by the reviewers.

---

## [Decision Letter · Decision Letter 2]

23 Aug 2024

PONE-D-23-37459R2Factors associated with knowledge and practice regarding oxygen administration: A cross-sectional study among registered nurses working in wards and ICUs at Muhimbili National Hospital in Dar es Salaam, TanzaniaPLOS ONE

Dear Dr. Ambikile,

Thank you for submitting your manuscript to PLOS ONE. After careful consideration, we feel that it has merit but does not fully meet PLOS ONE’s publication criteria as it currently stands. Therefore, we invite you to submit a revised version of the manuscript that addresses the points raised during the review process.

We look forward to receiving your revised manuscript.

Kind regards,

Agegnehu Bante

Academic Editor

PLOS ONE

Journal Requirements:

Reviewers' comments:

Reviewer's Responses to Questions

**Comments to the Author**

1. If the authors have adequately addressed your comments raised in a previous round of review and you feel that this manuscript is now acceptable for publication, you may indicate that here to bypass the “Comments to the Author” section, enter your conflict of interest statement in the “Confidential to Editor” section, and submit your "Accept" recommendation.

Reviewer #2: All comments have been addressed

Reviewer #4: All comments have been addressed

Reviewer #5: (No Response)

2. Is the manuscript technically sound, and do the data support the conclusions?

Reviewer #2: Yes

Reviewer #4: Yes

Reviewer #5: No

3. Has the statistical analysis been performed appropriately and rigorously? 

Reviewer #2: No

Reviewer #4: Yes

Reviewer #5: No

4. Have the authors made all data underlying the findings in their manuscript fully available?

Reviewer #2: (No Response)

Reviewer #4: Yes

Reviewer #5: No

5. Is the manuscript presented in an intelligible fashion and written in standard English?

Reviewer #2: Yes

Reviewer #4: Yes

Reviewer #5: No

6. Review Comments to the Author

Reviewer #2: (No Response)

Reviewer #4: General comment

This manuscript tried to explore Nurses’ knowledge and practices of oxygen administration in a referral hospital. The authors also examined the factors associated with these two outcomes. The authors did a good job of identifying this very critical aspect of healthcare that has not received a lot of attention. The manuscript is very well written, except for some minor corrections in the methods and results section.

Abstract

Well written

Introduction

Well written

Methods

Lines 122 to 125

The authors described the respondent selection process as “This technique ensured an equal chance of selecting the respondents. During the selection process, two pieces of paper marked as YES and NO were folded and placed in a box. Potential respondents were asked to pick one paper at a time without repeating. Those who selected a paper marked YES were included in the study.”

If only 2 pieces of paper were what respondents had to select from, then the authors did not ensure that each respondent had an equal chance of being selected. Please, state how many nurses were left after implementing the exclusion criteria and how each respondent was selected based on the principles of random probability, if that is what was done.

Lines 141 to 146

This statement is not clear. Kindly consider rephrasing.

Line 153

Kindly consider replacing ‘another’ with ‘a secondary’

Line 161

Kindly correct ‘in’

Lines 181 to 183

Kindly provide the Cronbach alpha value for the reliability test.

Line 195

Indicate the level of confidence used.

Results

Table 2

Write all abbreviations in full or clearly define them in the footnote, including ICU etc

Table 3

Correct ‘O2’ to O with a subscript 2

Discussion

Well written

Wrongly spelt words

There are a few words that need to be corrected in this manuscript. Check ‘incuding’ on line 281 and others

Thank you for the opportunity and good luck to the Authors.

Reviewer #5: Dear respected editor

Dear esteemed authors

Thank you for this opportunity to review manuscript titled “Factors associated with knowledge and practice regarding oxygen administration: A cross-sectional study among registered nurses working in wards and ICUs at Muhimbili National Hospital in Dar es Salaam, Tanzania” which was submitted to PLOS ONE

In general, the topic is required while the manuscript require major revisions

The manuscript requires English revision and editing using professional editing tools

Abstract:

- Abstract does not give readers practical implications from your study

- No need for the following statement “Data were then entered into an excel sheet and 25 transferred into the 25th version of IBM SPSS software for analysis” in method

- What about the instrument used in data collection? This should be reflected in method

Introduction:

- Background is very weak. Discuss the context of Tanzania in the light of international studies

- Develop a through literature review for your topic

- Generate research gap clearly and in depth, give the reader why you select oxygen administration to test with in your context.

- Discuss the significance of your study

- Use thought provoking statements that keep attention of readers

- You clearly state your aim of study in separate title

Method:

- Sampling is questionable. Give details about sampling selection and parameters of estimating sample size. Please revise sampling approach as what you mentioned is not random sampling

- - discuss clearly how you develop the study questionnaire. You develop new tool this go through certain steps. I recommend using devellis’ model for scale development otherwise, your study seems that you develop a test for students then you conclude results.

- What about the index of content validity of the newly developed tool?

- What about the construct validity of newly developed tool? This requires exploratory and confirmatory factor analysis.

- Also, The Cronbach's alpha is not enough to ensure reliability of newly developed tools, so you need to conduct test retest or item dimension correlations and interdimensional correlations

- What about common method bias? You need to use Harman’s one-factor test

- How do you screen respondents’ response bias or misconduct?

Results:

- On what base do you build regression? You need to conduct preliminary statistics that ensure if you need regression or not as a nova

Discussion:

It does not follow guidelines of effective debate

Despite you writing lengthy statements about implications, I cannot find what you conclude from your study. What are the clinical implications from your study I see general implications that could be written without research.

7. PLOS authors have the option to publish the peer review history of their article (what does this mean?). If published, this will include your full peer review and any attached files.

Reviewer #2: No

Reviewer #4: **Yes: **Addae Yaw Hammond

Reviewer #5: No

---

## [Author Response · Author response to Decision Letter 2]

11 Oct 2024

Response to reviewers and other comments have been uploaded as a file

---

## [Decision Letter · Decision Letter 3]

27 Nov 2024

PONE-D-23-37459R3Factors associated with knowledge and practice regarding oxygen administration: A cross-sectional study among registered nurses working in wards and ICUs at Muhimbili National Hospital in Dar es Salaam, TanzaniaPLOS ONE

Dear Dr. Ambikile,

Thank you for submitting your manuscript to PLOS ONE. After careful consideration, we feel that it has merit but does not fully meet PLOS ONE’s publication criteria as it currently stands. Therefore, we invite you to submit a revised version of the manuscript that addresses the points raised during the review process.

The comments raised in the previous version were not appropriately addressed. Work on them.

Please submit your revised manuscript by Jan 11 2025 11:59PM. If you will need more time than this to complete your revisions, please reply to this message or contact the journal office at plosone@plos.org. Please include the following items when submitting your revised manuscript:A rebuttal letter that responds to each point raised by the academic editor and reviewer(s). You should upload this letter as a separate file labeled 'Response to Reviewers'.A marked-up copy of your manuscript that highlights changes made to the original version. You should upload this as a separate file labeled 'Revised Manuscript with Track Changes'.An unmarked version of your revised paper without tracked changes. You should upload this as a separate file labeled 'Manuscript'.If applicable, we recommend that you deposit your laboratory protocols in protocols.io to enhance the reproducibility of your results. Protocols.io assigns your protocol its own identifier (DOI) so that it can be cited independently in the future. For instructions see: https://journals.plos.org/plosone/s/submission-guidelines#loc-laboratory-protocols. Additionally, PLOS ONE offers an option for publishing peer-reviewed Lab Protocol articles, which describe protocols hosted on protocols.io. Read more information on sharing protocols at https://plos.org/protocols?utm_medium=editorial-email&utm_source=authorletters&utm_campaign=protocols.

We look forward to receiving your revised manuscript.

Kind regards,

Agegnehu Bante

Academic Editor

PLOS ONE

Journal Requirements:

Additional Editor Comments:

Reviewers' comments:

Reviewer's Responses to Questions

**Comments to the Author**

1. If the authors have adequately addressed your comments raised in a previous round of review and you feel that this manuscript is now acceptable for publication, you may indicate that here to bypass the “Comments to the Author” section, enter your conflict of interest statement in the “Confidential to Editor” section, and submit your "Accept" recommendation.

Reviewer #4: All comments have been addressed

Reviewer #5: (No Response)

2. Is the manuscript technically sound, and do the data support the conclusions?

Reviewer #4: Yes

Reviewer #5: No

3. Has the statistical analysis been performed appropriately and rigorously? 

Reviewer #4: Yes

Reviewer #5: No

4. Have the authors made all data underlying the findings in their manuscript fully available?

Reviewer #4: Yes

Reviewer #5: No

5. Is the manuscript presented in an intelligible fashion and written in standard English?

Reviewer #4: Yes

Reviewer #5: No

6. Review Comments to the Author

Reviewer #4: The authors have adequately addressed all comments raised by the reviewer. The authors should, however, look through literature for the correct symbol for oxygen.

Reviewer #5: Dear respected editor 

Respected authors

Despite you making some amendments in this manuscript, the scientific rigor of the study is still questionable.

You should tell how you adapt your tools. How do you ensure the validity and reliability of study tools?

How do you control common method bias?. I gave you recommendations that could align this section with scientific principles of research; please see my previous recommendations. 

thanks 

Reviewer 5

7. PLOS authors have the option to publish the peer review history of their article (what does this mean?). If published, this will include your full peer review and any attached files.

Reviewer #4: No

Reviewer #5: No

---

## [Author Response · Author response to Decision Letter 3]

17 Dec 2024

Response to the reviewers' and editor's comments have been uploaded as a file.

---

## [Decision Letter · Decision Letter 4]

2 Jan 2025

Factors associated with knowledge and practice regarding oxygen administration: A cross-sectional study among registered nurses working in wards and ICUs at Muhimbili National Hospital in Dar es Salaam, Tanzania

PONE-D-23-37459R4

Dear Dr. Ambikie,

We’re pleased to inform you that your manuscript has been judged scientifically suitable for publication and will be formally accepted for publication once it meets all outstanding technical requirements.

Kind regards,

Agegnehu Bante

Academic Editor

PLOS ONE

Additional Editor Comments (optional):

Reviewers' comments:

Reviewer's Responses to Questions

**Comments to the Author**

1. If the authors have adequately addressed your comments raised in a previous round of review and you feel that this manuscript is now acceptable for publication, you may indicate that here to bypass the “Comments to the Author” section, enter your conflict of interest statement in the “Confidential to Editor” section, and submit your "Accept" recommendation.

Reviewer #5: All comments have been addressed

2. Is the manuscript technically sound, and do the data support the conclusions?

Reviewer #5: Yes

3. Has the statistical analysis been performed appropriately and rigorously? 

Reviewer #5: Yes

4. Have the authors made all data underlying the findings in their manuscript fully available?

Reviewer #5: Yes

5. Is the manuscript presented in an intelligible fashion and written in standard English?

Reviewer #5: Yes

6. Review Comments to the Author

Reviewer #5: Dear respected authors

Thank you for your efforts.

Please, in your incoming studies, consider adapting study tools to your context. It is important to ensure that tools have the same attributes in your context as the primary context where original tools were developed.

This point is still questionable in this study. However, I recommend publication of this study in its current form after referring to this point in the study limitations.

Best wishes

7. PLOS authors have the option to publish the peer review history of their article (what does this mean?). If published, this will include your full peer review and any attached files.

Reviewer #5: No

---

## [Editor Report · Acceptance letter]

7 Jan 2025

PONE-D-23-37459R4 

PLOS ONE

Dear Dr. Ambikile, 

I'm pleased to inform you that your manuscript has been deemed suitable for publication in PLOS ONE. Congratulations! Your manuscript is now being handed over to our production team.

Kind regards, 

on behalf of

Mr. Agegnehu Bante 

Academic Editor

PLOS ONE